# Correlations between single nucleotide polymorphisms in *FABP4* and meat quality and lipid metabolism gene expression in Yanbian yellow cattle

**Bao-zhen Yin**[1], **Jia-chen Fang**[2], **Jia-su Zhang**[1], **Luo-meng Zhang**[1], **Chang Xu**[1], **Hong-yan Xu**[1], **Jing Shao**[1], **Guang-jun Xia**[1] *

**1** Agricultural College of Yanbian University, Jilin Province, China, **2** Faculty of Agriculture and Life Science, Hirosaki University, Hirosaki, Japan

* ybuac@ybu.edu.cn

## Abstract

*FABP4* is a candidate gene for carcass and meat quality traits in livestock and poultry. However, the effects of *FABP4* have not been examined in the Yanbian yellow cattle, an economically important local cattle breed in China. In this study, we characterized single nucleotide polymorphisms (SNPs) in *FABP4* in this cattle breed and their associations with meat quality traits. Six SNPs (referred to as SNP1–6) were identified in *FABP4* by direct sequencing and polymerase chain reaction-restriction fragment length polymorphism. The six SNPs were significantly correlated with meat quality traits. In particular, the GG and GA genotypes of SNP1 were significantly associated with water and fat contents and GG and AA genotypes of SNP1 were significantly associated with protein contents ($P < 0.05$). The fat content and marbling in heterozygous individuals at SNP2–6 were significantly higher than those in wild-type or mutant individuals ($P < 0.05$), while protein content was significantly higher in wild-type and mutant individuals than in heterozygous individuals ($P < 0.05$). A gene expression analysis indicated that the lipid metabolism-related genes *FABP4*, *PPARγ*, *ANGPTL4*, and *LPL* show similar expression patterns with respect to *FABP4* genotypes, with the highest levels in wild-type individuals and the lowest levels in mutants. In conclusion, *FABP4* SNPs can be used for marker-assisted selection in Yanbian yellow cattle breeding.

## Introduction

In 1972, Ockner et al. [1] first discovered the fatty acid binding protein (*FABP*) family in a study of the intestinal mucosa and other tissues in animals, and fatty acid binding protein 4 (*FABP4* or *A-FABP*), a *FABP* family member, was discovered by Spiegelman et al. in 1983 [2]. Damon et al. [3] found that levels of the protein encoded by porcine *FABP4* are positively related to the fat cell count and lipid content. Yan Wei et al. [4] found that *FABP4 c.246 + 37A>G* and *c.348 + 298T>C* in Chinese and New Zealand sheep populations are potential

**Funding:** The research was supported by Key scientific and technological projects of Jilin Provincial Science and technology development plan (20160204017NY); scientific and technological projects of the 13th five year plan of Jilin Provincial Department of Education (JJKH20180903KJ).(http://kjt.jl.gov.cn/ http://jyt.jl.gov.cn/).The funders had no role in study design, data collection and analysis, decision to publish, or preparation of the manuscript.

**Competing interests:** The authors have declared that no competing interests exist.

molecular markers for intramuscular fat content in Tibetan sheep. Li Xiaoling et al. [5] evaluated the expression of *A-FABP* in the porcine heart, liver, spleen, lung, kidney, longissimus dorsi, and leg muscle tissues by quantitative real-time polymerase chain reaction (qRT-PCR) and found that levels *A-FABP* are the highest in the longissimus dorsi and leg muscle and lowest in the lung. Another study has shown that *FABP4*-E1-51 (the 51st site of exon 1) determines the sebum thickness and intramuscular fat content of Luqin 3; the sebum thickness is significantly greater for the CC and CT genotypes than the TT genotype, and the intramuscular fat content is significantly higher for the CC and TT genotypes than the CT genotype [6]. In the first exon of *FABP4*, a C→T mutation at position 51 significantly affects the intramuscular fat content in chicken [7]. The *g.2834C>G* polymorphism in *FABP4* in Qinchuan cattle is significantly correlated with meat quality traits and the eye muscle area [2], while *g.7516G<C* in cattle is significantly related to marbling and the intramuscular fat content [8].

Based on these findings, *FABP4* is a candidate gene associated with the meat quality characteristics of livestock and poultry. Yanbian yellow cattle are mainly reared in the Yanbian area, Jilin Province, China. It is one of the top five local cattle breeds in China, where it is a key protected and developing breed. This breed has excellent meat quality and substantial intramuscular fat deposition, and is therefore, indispensable for the beef cattle industry and for improving meat varieties in China [9–11]. However, the genetic determinants of meat quality of this breed are not well-characterized. We identified single nucleotide polymorphisms (SNPs) in *FABP4* of Yanbian yellow cattle by PCR-restriction fragment length polymorphism (PCR-RFLP) and sequencing and evaluated the relationships between polymorphisms and meat quality traits. In addition, qRT-PCR was used to compare expression levels of lipid metabolism-related genes in cattle with different *FABP4* genotypes. The results of our study clarify the effects of variations in *FABP4* on intramuscular lipid metabolism and provide a theoretical basis for breeding Yanbian yellow cattle with desirable meat quality traits.

## Materials and methods

### Test animals and sample collection

In total, 350 bulls of Yanbian yellow cattle were selected from the Yanbian Animal Husbandry Development Group Co., Ltd. of Jilin Province. Animal care and experiments were in accordance with the guidelines established by the Regulation for the Administration of Affairs Concerning Experimental Animals (Ministry of Science and Technology, China, 2004) and the study was approved by the Medical Ethics Committee, Affiliated Hospital of Yanbian University (Yanbian Hospital) (Approval ID: 201702). The cattle had free access to feed and fresh water. All cattle were fattened under the same feeding conditions and management conditions and slaughtered at 30 months of age. Feeding was stopped 24 h before slaughter, and a quiet environment and adequate drinking water were provided. Animals were euthanized by electric shock. Before slaughter, 25 mL of blood was collected from the jugular vein, anticoagulant was added, and samples were stored separately at –80˚C. The 12th to 13th intercostal longissimus dorsi muscle was collected and stored at –80˚C after vacuum packaging to determine meat quality, and 3 g of longissimus dorsi muscle tissue was collected and immediately stored in liquid nitrogen for total RNA extraction.

### Determination of traits

Meat quality was evaluated based on the water content, fat content, protein content, thickness of backfat, and marble pattern. The water content was determined by the direct drying method. The intramuscular fat content was determined according to Sox's extraction technique for a feed analysis and feed quality detection, as published by the Agricultural University

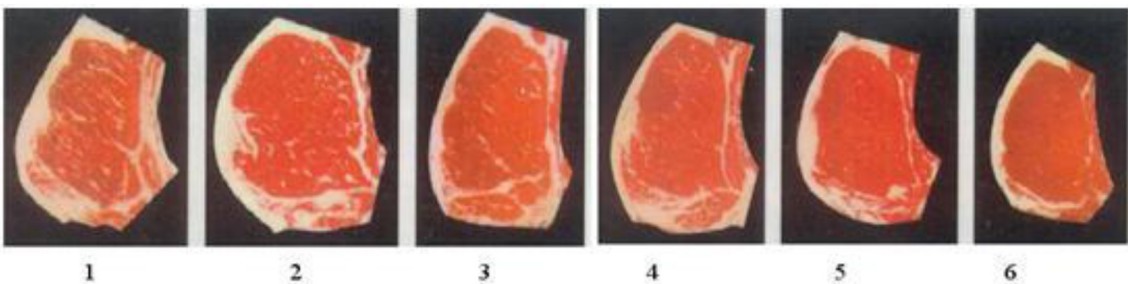

**Fig 1. Grade map of marbling.**

of China. The protein content in beef was determined by the KN method [12]. After slaughter, the thickness of subcutaneous fat in the longissimus dorsi (i.e., backfat thickness) was measured vertically using a helical micrometer. The transverse section of the longissimus dorsi was observed, and the marbling grade was determined by referring to the rating map shown in Fig 1 [13].

## Genomic DNA extraction, PCR amplification, sequencing, and PCR-RFLP

DNA samples of Yanbian yellow cattle were extracted according to the instructions provided with the Blood Genomic DNA Extraction Kit (0.1–1 mL) (DP318; Tiangen Biochemical Technology Co., Ltd., Beijing, China) (S1 File). The purity and concentration of each genomic DNA sample were detected using the NanoDrop 1000 spectrometer (Thermo Scientific, Waltham, MA, USA).

A pair of primers (F: 5′–ACCCCTATGATGCTATTCCACA–3′ and R: 5′–ATACGGTTCAC ATTGAGAGA–3′) was used to amplify the exon 3 of *FABP4* based on the bovine genome sequence (NCBI accession NC_007312.4) [14]. The final 565-bp amplicon was synthesized in a 20-μL PCR mixture containing 2 μL of DNA template, 0.5 μL of upstream and downstream primers, 7 μL of ddH$_2$O, and 10 μL of 2× Taq PCR master mix (TIANGEN). The cycling protocol consisted of denaturation for 5 min at 95˚C, followed by 30 cycles of 94˚C for 30 s, annealing for 30 s at 55˚C, primer extension at 72˚C for 30 s, and a final extension at 72˚C for 10 min. The PCR products were detected by 1% agarose gel electrophoresis and sent to Shenggong Biotechnology Co., Ltd. (Shanghai, China) for sequencing.

Six SNPs were found in the third exon of *FABP4* by DNA sequencing, and SNP1 was found at the *Nla*III restriction site. Therefore, SNP1 was genotyped by PCR-RFLP and SNP2–6 were analyzed based on the sequencing results. The PCR product (10 μL) was digested with *Nla*III (New England Biolabs (Beijing) Ltd., Beijing, China) for 2 h at 37˚C, and the reaction was stopped by heating at 65˚C for 20 min. The digested products were detected by 1% agarose gel electrophoresis.

## Total RNA extraction and cDNA synthesis

Total RNA was extracted according to the instructions provided with the Eastep Super Total RNA Extraction Kit (LS1040) of Pleuger Biological Products Co., Ltd. (Shanghai, China). RNA was quantified at 260 nm by ultraviolet spectrophotometry. The A$_{260}$/A$_{280}$ values for all RNAs ranged from 1.8 to 2.1. The integrity of RNA samples was detected by 1% agarose gel electrophoresis. The extracted RNA produced clear 28S and 18S rRNA bands during denaturing gel electrophoresis. The 28S rRNA band was approximately twice as intense as the 18S rRNA band. The total volume of the cDNA synthesis reaction was 20 μL, containing 4 μL of 5×

PrimeScript RT Master Mix, less than 500 ng of total RNA, and RNase-free ddH$_2$O. The reaction procedure included reverse transcription at 37˚C for 15 min, inactivation at 85˚C for 5 s, and holding at 4˚C.

## Expression analysis by qRT-PCR

The primers were designed with reference to the sequences of bovine *FABP4* (NM_001114667.1), *PPARγ* (NM_181024), *ANGPTL4* (NM_001046043.2), *LPL*, and *GAPDH* (BC102589) in NCBI (Table 1), and *GAPDH* was selected as the internal reference [15]. PCR was performed using the SYBR Premix Ex Taq II (Tli RNaseH Plus) Kit, following the manufacturer's instructions. The reaction volume was 20 μL, including 10 μL of SYBR Premix Ex Taq II (Tli RNaseH Plus) (2×), 0.8 μL each of forward and reverse primers (10 μM each), 2 μL of cDNA, and 6.4 μL of RNase-free ddH$_2$O. The reaction procedure was as follows: (1) predenaturation at 95˚C for 30 s, 1 cycle; (2) PCR (analysis mode: quantitative) at 95˚C for 5 s and 60˚C for 30 s, 40 cycles; (3) melting (analysis mode: melting curve) at 95˚C for 5 s, 60˚C for 1 min, and 95˚C, 1 cycle; (4) cooling at 50˚C for 30 s, 1 cycle.

## Statistical analysis

Genotype and allele frequencies were directly calculated for all SNPs from sequence alignments. Hardy–Weinberg disequilibrium ($\chi^2$), gene heterozygosity ($H_e$), and effective numbers of alleles ($N_e$) were calculated according to previously described approaches [16,17]. The polymorphism information content (PIC) was calculated based on Botstein's method [18].

One-way analysis of variance (ANOVA) was used to evaluate the associations of SNPs with meat traits using SPSS 20. The data are shown as the means ± standard deviations; $P < 0.05$ was considered significant.

To determine the probability of recombination between *FABP4* sites, SHEsis was used to analyze linkage disequilibrium (LD; estimated by the parameter $r^2$) [19]. The relative expression levels of lipid metabolism genes were calculated by the $2^{-\Delta\Delta CT}$ method, in which $\Delta CT = CT_{target\ gene} - CT_{internal\ reference\ gene}$ and $\Delta_{\Delta CT} = \Delta CT_{test\ group} - \Delta CT_{control\ group}$.

## Results

### Genetic variation in *FABP4* in Yanbian yellow cattle

Gel electrophoresis results for genomic DNA obtained from the blood samples are shown in S1 Fig. Amplification results for the *FABP4* gene are shown in Fig 2. Based on a sequence analysis, we detected the following six SNPs in exon 3 (Figs 3–8): SNP1, *g.3691G>A*; SNP2,

**Table 1. Primers for the analysis of *FABP4* gene expression in Yanbian yellow cattle.**

| Gene | Primer sequences (5'–3') | Product size | Annealing temperature (˚C) |
|---|---|---|---|
| *FABP4* | F: CAGTGTAAATGGGGATGTGG | 264 bp | 60 |
| | R: CTCTCGTAAACTCTGGTAGC | | |
| *PPARγ* | F: CCTTCACCACCGTTGACTTCTC | 145 bp | 60 |
| | R: GATACAGGCTCCACTTTGATTC | | |
| *ANGPTL4* | F: GATGGCTCCGTGGACTTTAACC | 103 bp | 60 |
| | R: GGATGTGATGCACCTTCTCCAG | | |
| *LPL* | F: AGTGCCTGCTTGTTTGTG | 286 bp | 60 |
| | R: TATGCCCTTTCTGTTCCT | | |
| *GAPDH* | F: ACCCAGAAGACTGTGGATGG | 247 bp | 60 |
| | R: ACGCCTGCTTCACCACCTTC | | |

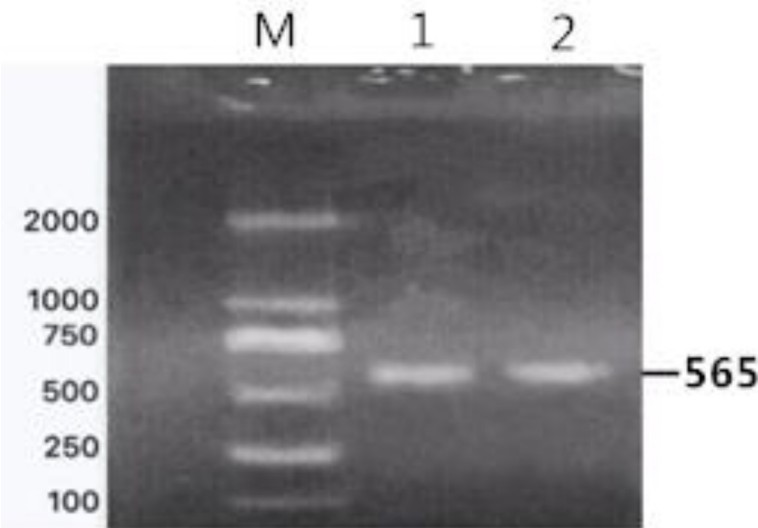

**Fig 2. Electrophoretic map of the amplification products of *FABP4* in Yanbian yellow cattle.**

*g.3496A>C*; SNP3, *g.3533A>T*; SNP4, *g.3711G>C*; SNP5, *g.3745T>C*; and SNP6, *g.3767T>C*. At SNP1, digestion of the 565-bp PCR fragment of *FABP4* with *Nla*III resulted in fragment lengths of 469, 236, and 233 bp for genotype GA, 236 and 233 bp for genotype AA, and 469 bp for genotype GG (Fig 9).

Based on analyses of genotype and allele frequencies (Table 2), we found that the six SNPs in *FABP4* exhibited intermediate levels of diversity (0.25 < PIC < 0.50). SNP1 was in Hardy–Weinberg equilibrium. For SNP1, the GG genotype (68.57%) was the most frequent, followed by GA (18.57%) and AA (12.86%). The allele frequencies were 0.7786 (G) and 0.2214 (A). Homozygosity ($H_o$), heterozygosity ($H_e$), and the effective allele number ($N_e$) were 0.6522, 0.3448, and 1.5263, respectively. The AC genotype (48.57%) of SNP2 was the most frequent, followed by AA (32.86%) and CC (18.57%). The allele frequencies were 0.5715 (A) and 0.4285 (C). The $H_o$, $H_e$, and $N_e$ values for the locus were 00.5102, 0.4898, and 1.9600, respectively. The AT genotype (47.14%) of SNP3 was the most frequent, followed by AA (35.72%) and TT (17.14%). The allele frequencies were 5929 (A) and 0.4071 (T). The $H_o$, $H_e$, and $N_e$ values were 0.5173, 0.4827, and 1.9331, respectively. The GG genotype (44.29%) of SNP4 was the most

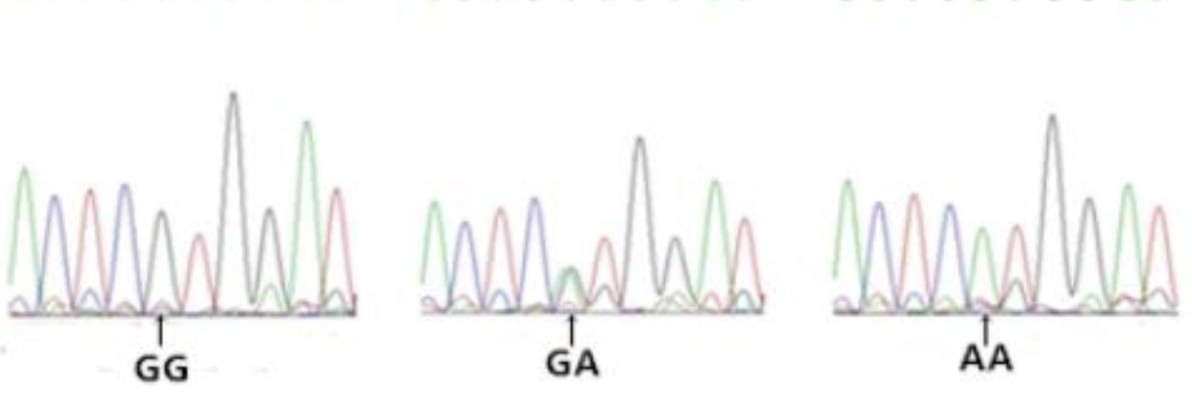

**Fig 3. Sequencing map of the novel SNP1 in *FABP4*.**

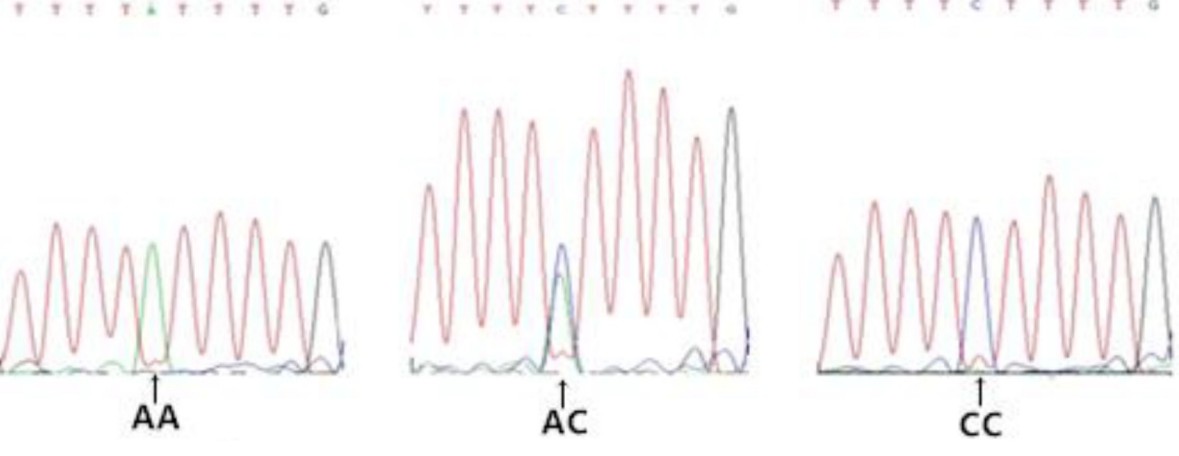

**Fig 4. Sequencing map of the novel SNP2 in *FABP4*.**

frequent, followed by GC (38.57%) and CC (17.14%). The allele frequencies were 0.6358 (G) and 0.3642 (C). The $H_o$, $H_e$, and $N_e$ values were 0.5369, 0.4631, and 1.8625, respectively. The TT genotype (44.29%) of SNP5 was the most frequent, followed by TC (38.57%) and CC (17.14%). The allele frequencies were 0.6358 (T) and 0.3642 (C). The $H_o$, $H_e$, and $N_e$ values were 00.5369, 0.4631, and 1.8625, respectively. The TC genotype (47.14%) of SNP6 was the most frequent, followed by TT (35.72%) and CC (17.14%). The allele frequencies were 0.5929 (T) and 0.4071 (C). The $H_o$, $H_e$, and $N_e$ values were 0.5173, 0.4827, and 1.9331, respectively.

## Linkage disequilibrium analysis

For SNP2, SNP3, SNP4, SNP5, and SNP6 in *FABP4*, LD was analyzed using SHEsis. The results of this analysis are summarized in Table 3 and Fig 10. Five of the SNPs were in strong LD, with similar genetic effects ($r^2 > 0.33$). Among these, SNP3 and SNP6 as well as SNP4 and SNP5 were in complete LD ($r^2 = 1$). All marker information could be obtained by observing a single site.

## Effect of *FABP4* polymorphisms on meat quality

Associations between *FABP4* polymorphisms and meat quality traits, including meat water content, fat content, protein content, before-slaughter weight, body weight, and backfat

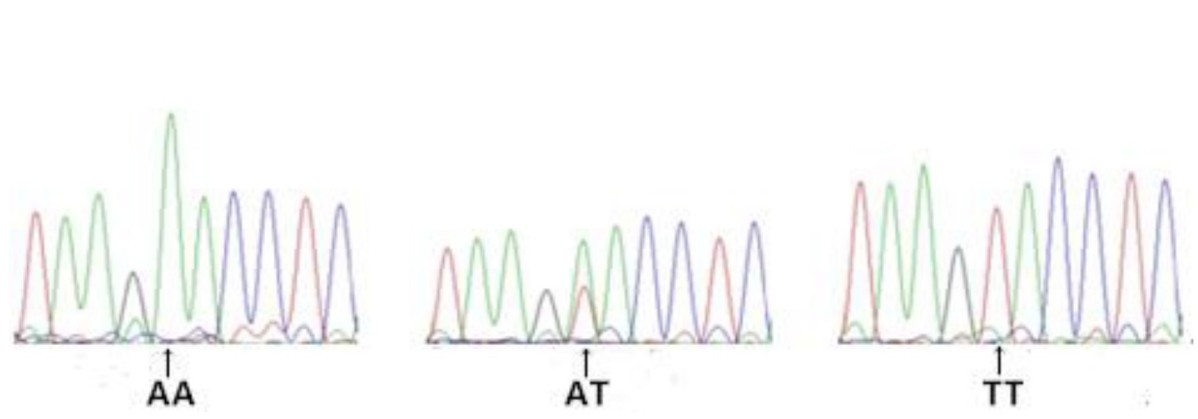

**Fig 5. Sequencing map of the novel SNP3 in *FABP4*.**

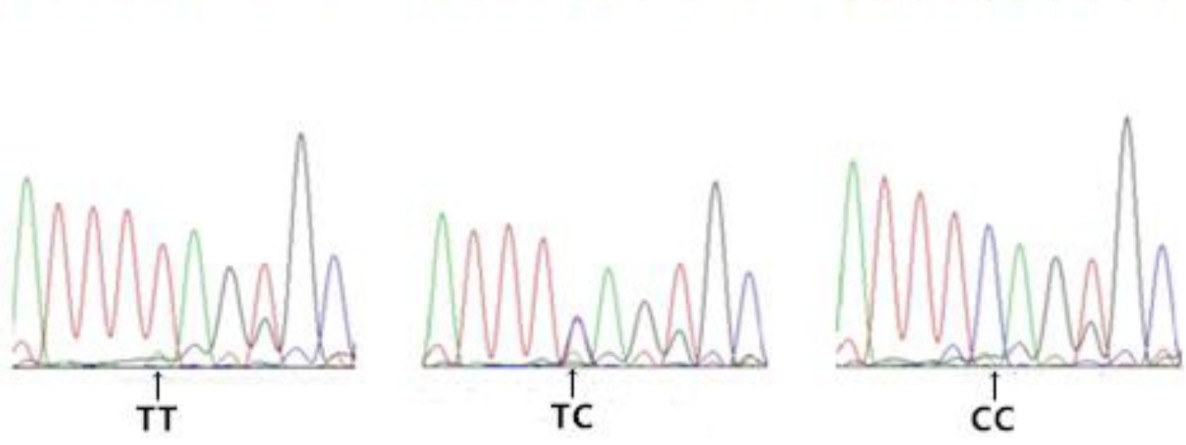

**Fig 6. Sequencing map of the novel SNP4 in *FABP4*.**

thickness in Yanbian yellow cattle were evaluated (Table 4). The water content was significantly higher for the GG genotype at SNP1 than for the GA and AA genotypes *(P < 0.05)*. The fat content was significantly higher for the GA genotype than for the GG and AA genotypes *(P < 0.05)*. The protein content was significantly higher for the GG and AA genotypes than for the GA genotype *(P < 0.05)*. The fat contents of heterozygous individuals at SNP2, SNP3, SNP4, SNP5, and SNP6 were significantly higher than those of wild-type individuals and mutants *(P < 0.05)*. The protein contents of wild-type individuals and mutants were significantly higher than those of heterozygous individuals *(P < 0.05)*. The marbling pattern of heterozygous individuals was significantly superior to that of mutants *(P < 0.05)*.

### Expression of fat metabolism genes in Yanbian yellow cattle with different *FABP4* genotypes

Gel electrophoresis results for extracted RNA are shown in S2 Fig. *FABP4, PPARγ, ANGPTL4,* and *LPL* encode key enzymes involved in fat synthesis and decomposition [20,21]. The main function of *LPL* is to catalyze the hydrolysis of triglycerides to produce glycerol and free fatty acids, thereby providing energy to tissues, or to re-esterify triglycerides for storage in adipose

**Fig 7. Sequencing map of the novel SNP5 in *FABP4*.**

T A A T T G T T A T T A A T T G T T A T T A A T C G T T A T

TT TC CC

**Fig 8. Sequencing map of the novel SNP6 in *FABP4*.**

tissue [22,23]. *PPARγ* can induce the formation of small fat cells and regulate the expression of *LPL*, *FABP4*, and other genes [24]. *ANGPTL4* increases the triglyceride content and promotes fat deposition by inhibiting the activity of *LPL* [25]. The expression levels of four lipid metabolism genes for different *FABP4* genotypes in Yanbian yellow cattle are summarized in Fig 11 and Table 5. The expression levels of *FABP4* and *LPL* were significantly higher in individuals with wild-type *FABP4* than in heterozygous and homozygous mutants ($P < 0.01$). The expression levels of *PPARγ* were significantly higher in wild-type individuals than in mutants ($P < 0.05$), and levels of *ANGPTL4* were significantly higher in wild-type individuals than in heterozygous and homozygous mutants ($P < 0.05$).

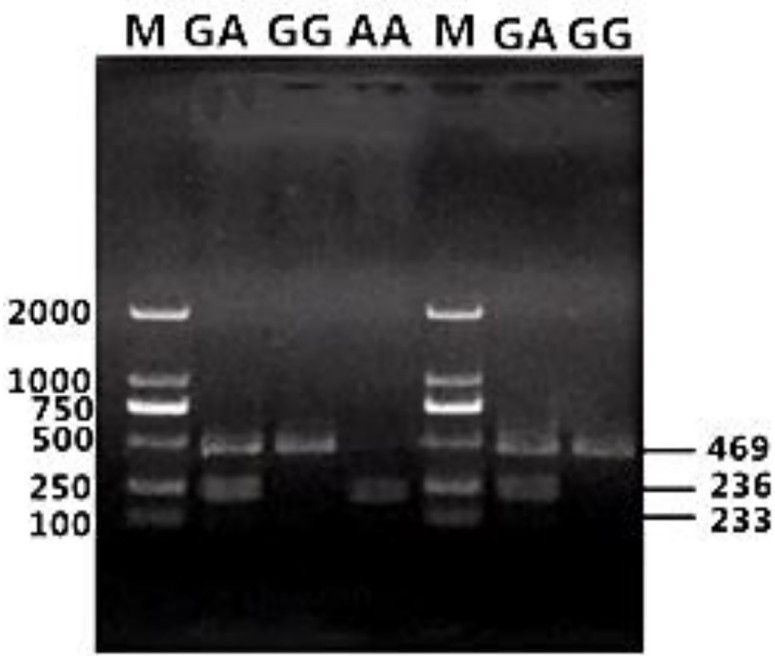

**Fig 9. PCR-RFLP results for *FABP4* (SNP1).**

**Table 2. Summary of genetic variation for different sites of *FABP4* in Yanbian yellow cattle.**

| Site | SNP1 | SNP2 | SNP3 | SNP4 | SNP5 | SNP6 |
|---|---|---|---|---|---|---|
| Allele frequency | G-0.7786 | A-0.5715 | A-0.5929 | G-0.6358 | T-0.6358 | T-0.5929 |
|  | A-0.2214 | C-0.4285 | T-0.4071 | C-0.3642 | C-0.3642 | C-0.4071 |
| Genotype frequency | GG-0.6857 | AA-0.3286 | AA-0.3572 | GG-0.4429 | TT-0.4429 | TT-0.3572 |
|  | GA-0.1857 | AC-0.4857 | AT-0.4714 | GC-0.3857 | TC-0.3857 | TC-0.4714 |
|  | AA-0.1286 | CC-0.1857 | TT-0.1714 | CC-0.1714 | CC-0.1714 | CC-0.1714 |
| $\chi^2$ | 48.9459 | 0.9959 | 1.0529 | 5.9235 | 5.9235 | 1.0529 |
| $H_o$ | 0.6552 | 0.5102 | 0.5173 | 0.5369 | 0.5369 | 0.5173 |
| $H_e$ | 0.3448 | 0.4898 | 0.4827 | 0.4631 | 0.4631 | 0.4827 |
| $N_e$ | 1.5263 | 1.96 | 1.9331 | 1.8625 | 1.8625 | 1.9331 |
| PIC | 0.2854 | 0.3699 | 0.3662 | 0.3559 | 0.3559 | 0.3662 |

$H_o$: Homozygosity; $H_e$: Heterozygosity; $N_e$: Number of effective alleles. $\chi^2$: Test of conformance to Hardy–Weinberg equilibrium ($P > 0.05$ indicates a state of genetic balance). $\chi^2_{0.05} = 5.991$, $\chi^2_{0.01} = 9.210$. PIC: polymorphism information content, evaluated by a chi squared test based on the observed and expected values.

## Discussion

### Relationships between SNPs in *FABP4* and meat quality traits in Yanbian yellow cattle

Kenji et al. [26] found that the *I74V* locus of *FABP4* in Japanese black cattle is significantly correlated with the palmitoleic acid and linoleic acid contents in intramuscular fat. Hengwei et al. [27] found that *g.2834C>G* of *FABP4* is significantly correlated with the eye muscle area and intramuscular fat content, and *g.4420A>G* is significantly correlated with backfat thickness. Cho et al. [28] found that *g.220A>G (I74v)* in exon 2 and *g.348+303T>C* in intron 3 of *FABP4* are significantly correlated with fatty acid deposition and backfat thickness in cattle. In the current study, there was no correlation between six SNPs in *FABP4* and back fat thickness in Yanbian yellow cattle. This difference among studies could be explained by the difference in exons examined. A previous study has shown that there is a significant correlation between the marbling pattern and *g.3631G>A* in *FABP4*, and between *g.3473A>T* and carcass weight [14]. Another study has reported that *g.3691G>A* in *FABP4* is significantly correlated with marbling and the meat quality score in Korean cattle [29]. The corresponding SNP in our study, SNP1 (*g.3691G>A*), was not correlated with marbling, but was significantly correlated with water, fat, and protein contents. The differences between studies may be explained by difference in sample sizes or genetic background. Additionally, SNP2, SNP3, SNP4, SNP5, and SNP6 were in LD and significantly influenced the fat and protein contents and marbling grade of Yanbian yellow cattle. In conclusion, *FABP4* is a candidate gene for improving beef quality traits in Yanbian yellow cattle.

**Table 3. Linkage disequilibrium in the *FABP4* gene.**

| $r^2$ | SNP3 | SNP4 | SNP5 | SNP6 |
|---|---|---|---|---|
| SNP2 | 0.944 | 0.791 | 0.791 | 0.944 |
| SNP3 | — | 0.838 | 0.838 | 1.000 |
| SNP4 | — | — | 1.000 | 0.838 |
| SNP5 | — | — | — | 0.838 |

LD was computed for all possible combinations of five SNPs as $r^2$ values.

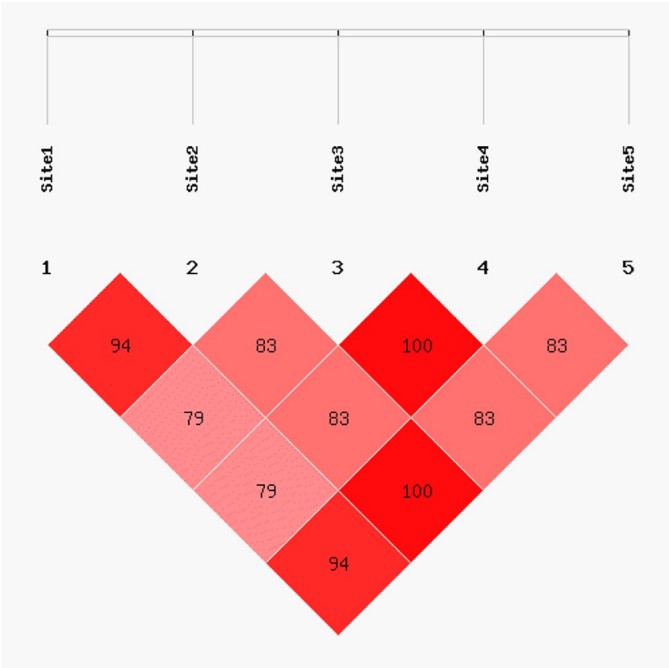

**Fig 10. Linkage disequilibrium of five SNPs in *FABP4*.**

## Expression of fat metabolism genes in Yanbian yellow cattle with different *FABP4* genotypes

*FABP4*, *PPARγ*, *ANGPTL4*, and *LPL* are important lipid metabolism genes. *LPL* and *PPARγ* had a positive regulatory effect on fat metabolism [30,31]. Chang et al. [11] showed that the expression of *ANGPTL4* in the bovine longissimus dorsi muscle is positively correlated with the intramuscular fat content. *FABP4* mRNA levels in Placental trophoblast cells increase in response to a *PPARγ* agonist, indicating that *PPARγ* can increase the expression of *FABP4* [32]. Xiao [33] found that the overexpression of *FABP4* could promote lipid deposition, significantly increase *PPARγ* expression, and inhibit *LPL* expression.

In the longissimus dorsi muscle of Nanjiang yellow sheep, the levels of *FABP4* were correlated with the levels of *PPARγ2* [34]. Additionally, the expression levels of four genes, *FABP4*, *PPARγ*, *ANGPTL4*, and *LPL*, were the highest in individuals with wild-type *FABP4* and lowest in homozygous mutants. Consistent with these findings, in our study, the expression levels of the four genes were significantly higher in individuals with wild-type *FABP4* than in heterozygous and/or homozygous mutants. This was a preliminary analysis of lipid metabolism genes (i.e., *FABP4*, *PPARγ*, *ANGPTL4*, and *LPL*) with respect to genotypes at five SNPs in LD in *FABP4* in Yanbian yellow cattle. In the future, studies with larger sample sizes are needed to evaluate associations between protein levels and the activity of intramuscular fat cells under the same conditions to establish a concrete theoretical basis for the improvement of beef quality in Yanbian yellow cattle.

## Conclusions

Six SNPs with moderate variation in Hardy-Weinberg equilibrium were found in exon 3 of *FABP4* in Yanbian yellow cattle. Among them, SNP1 was significantly correlated with the

**Table 4. Correlation between SNPs in *FABP4* and meat quality traits of Yanbian yellow cattle.**

| Site | Moisture (%) | Fat (%) | Protein (%) | Live weight before slaughter (kg) | Carcass weight (kg) | Backfat thickness (cm) | Marbling (grade) |
|------|------|------|------|------|------|------|------|
| SNP1 | | | | | | | |
| GG | 56.67±0.69[a] | 13.14±1.65[b] | 21.60±1.44[a] | 559.00±18.10 | 315.90±11.53 | 1.30±0.13 | 3.00±0.71 |
| GA | 53.00±2.52[b] | 26.00±6.11[a] | 15.00±1.15[b] | 600.00±47.25 | 327.00±28.75 | 1.50±0.50 | 3.00±0.58 |
| AA | 53.00±0.58[b] | 14.00±2.08[b] | 20.67±1.76[a] | 606.67±28.48 | 336.00±29.14 | 1.60±0.40 | 2.50±0.76 |
| SNP2 | | | | | | | |
| AA | 55.60±1.47 | 13.75±1.25[b] | 22.33±1.67[a] | 564.00±36.55 | 318.40±23.81 | 1.50±0.27 | 3.00±0.41[ab] |
| AC | 52.80±1.98 | 25.50±5.84[a] | 15.25±1.31[b] | 585.00±23.98 | 337.60±12.08 | 1.60±0.19 | 2.00±0.40[b] |
| CC | 56.00±0.68 | 13.25±1.03[b] | 22.75±1.89[a] | 565.00±18.75 | 317.17±13.57 | 1.17±0.21 | 4.00±0.26[a] |
| SNP3 | | | | | | | |
| AA | 54.67±1.52 | 13.20±1.11[b] | 23.00±1.73[a] | 581.67±34.68 | 322.00±19.77 | 1.58±0.24 | 2.67±0.88[ab] |
| AT | 53.50±2.40 | 25.00±6.26[a] | 14.67±0.33[b] | 590.00±22.73 | 337.00±15.57 | 1.50±0.20 | 2.00±0.59[b] |
| TT | 56.20±0.80 | 11.50±1.44[b] | 21.67±2.19[a] | 562.00±22.67 | 318.00±16.59 | 1.20±0.25 | 4.25±0.48[a] |
| SNP4 | | | | | | | |
| GG | 54.57±1.29 | 13.33±0.92[b] | 23.20±1.36[a] | 582.86±29.34 | 324.29±16.87 | 1.57±0.20 | 2.50±0.65[ab] |
| GC | 54.40±2.06 | 24.40±4.88[a] | 15.75±1.11[b] | 598.00±19.34 | 339.20±12.26 | 1.50±0.16 | 1.75±0.48[b] |
| CC | 56.17±0.65 | 11.40±1.12[b] | 22.75±1.89[a] | 573.33±21.71 | 326.50±15.99 | 1.33±0.25 | 4.00±0.41[a] |
| SNP5 | | | | | | | |
| TT | 54.29±1.34 | 13.17±0.91[b] | 23.20±1.36[a] | 590.00±30.47 | 328.86±18.06 | 1.64±0.21 | 2.20±0.58[ab] |
| TC | 52.40±2.16 | 23.25±5.20[a] | 15.50±1.19[b] | 578.00±21.31 | 328.60±14.70 | 1.40±0.19 | 2.00±0.41[b] |
| CC | 56.17±0.65 | 11.60±1.12[b] | 21.25±1.03[a] | 565.00±18.75 | 319.67±13.65 | 1.25±0.21 | 3.75±0.48[a] |
| SNP6 | | | | | | | |
| TT | 54.80±1.85 | 13.75±1.25[b] | 24.67±0.67[a] | 572.00±40.79 | 314.40±22.36 | 1.50±0.27 | 2.33±0.88[ab] |
| TC | 52.33±2.96 | 25.50±5.84[a] | 16.00±1.53[b] | 593.33±31.80 | 337.33±22.02 | 1.50±0.29 | 2.00±0.58[b] |
| CC | 56.75±0.75 | 11.50±1.44[b] | 23.00±1.73[a] | 565.00±29.01 | 317.75±21.42 | 1.38±0.24 | 4.25±0.48[a] |

Within a column, different lowercase letters indicate a significant difference ($P < 0.05$), while identical lowercase letters or a lack of superscript letters indicates a non-significant difference ($P > 0.05$).

water, fat, and protein contents of Yanbian yellow cattle. SNP2, SNP3, SNP4, SNP5, and SNP6 were in LD, and SNP3 and SNP6, as well as SNP4 and SNP5, were in perfect LD ($r^2 = 1$). At

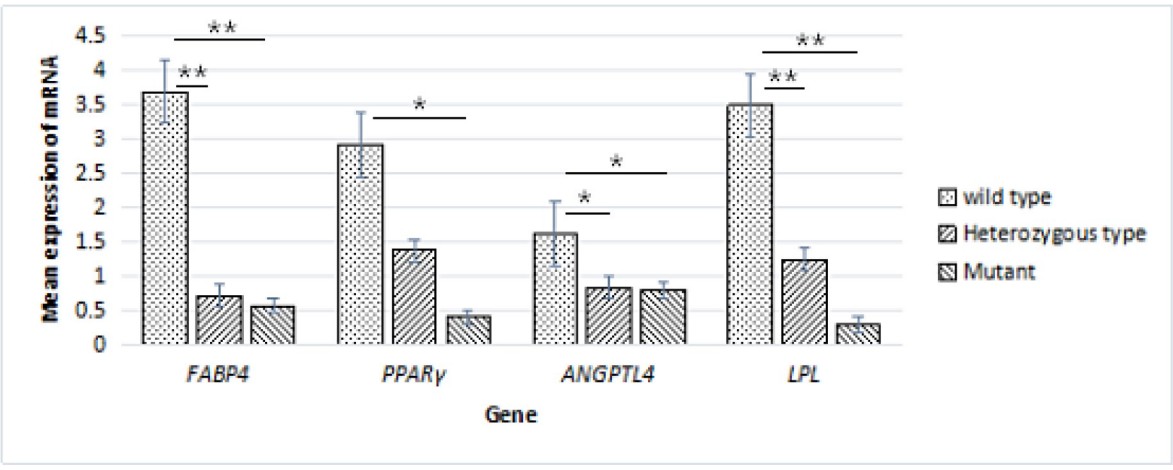

**Fig 11. Expression levels of lipid metabolism genes with respect to *FABP4* genotype in Yanbian yellow cattle.**

**Table 5. Expression levels of *FABP4*, *PPARγ*, *ANGPTL4*, and *LPL* with respect to the *FABP4* genotype.**

| Gene | Wild-type | Heterozygous | Mutants |
|---|---|---|---|
| *FABP4* | 3.69±0.73[A] | 0.71±0.28[B] | 0.57±0.21[B] |
| *PPARγ* | 2.92±1.09[a] | 1.38±0.32[ab] | 0.40±0.18[b] |
| *ANGPTL4* | 1.62±0.25[a] | 0.83±0.10[b] | 0.80±0.10[b] |
| *LPL* | 3.49±0.68[A] | 1.24±0.47[B] | 0.30±0.12[B] |

Values with different superscript letters (a, b) within the same row differ significantly at $P < 0.05$. Values with different superscript letters (A, B) within the same row differ significantly at $P < 0.01$.

these five loci, the wild-type allele was dominant. These SNPs significantly affected the fat content, protein content, and marbling grade of Yanbian yellow cattle. Four lipid metabolism genes, *FABP4*, *PPARγ*, *ANGPTL4*, and *LPL*, had similar expression patterns in animals with different *FABP4* genotypes, with the highest levels in wild-type individuals and the lowest levels in mutant individuals. Based on these findings, *FABP4* is a candidate gene affecting fat metabolism in cattle, and the SNPs identified in this study can be used as molecular markers for cattle breeding.

## Supporting information

**S1 Fig. Agarose gel electrophoresis.**
(DOCX)

**S2 Fig. Agarose gel electrophoresis of RNA.**
(DOCX)

**S3 Fig. Un-cropped images of electrophoretic map analyses shown in Fig 2.**
(DOCX)

**S4 Fig. Un-cropped images of electrophoretic map analyses shown in Fig 9.**
(DOCX)

**S5 Fig. Un-cropped images of electrophoretic map analyses shown in S1 Fig.**
(DOCX)

**S6 Fig. Un-cropped images of electrophoretic map analyses shown in S2 Fig.**
(DOCX)

**S1 File. Procedure for the Blood Genomic DNA Extraction Kit.**
(DOC)

## Acknowledgments

We would like to thank Dr. Lichun Zhang for help with statistical analyses. We would like to thank Editage (www.editage.cn) for their assistance with English language editing.

## Author Contributions

**Conceptualization:** Bao-zhen Yin, Jia-chen Fang.

**Formal analysis:** Chang Xu, Jing Shao.

**Funding acquisition:** Hong-yan Xu, Guang-jun Xia.

**Investigation:** Bao-zhen Yin, Jia-su Zhang, Luo-meng Zhang, Chang Xu, Jing Shao.

**Methodology:** Bao-zhen Yin, Jia-chen Fang, Guang-jun Xia.

**Resources:** Hong-yan Xu, Guang-jun Xia.

**Supervision:** Jia-su Zhang, Luo-meng Zhang.

**Visualization:** Bao-zhen Yin.

**Writing – original draft:** Bao-zhen Yin.

**Writing – review & editing:** Bao-zhen Yin, Hong-yan Xu, Guang-jun Xia.

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
