## [Decision Letter · Decision Letter 0]

24 Mar 2020

PONE-D-20-01041

Correlation between single nucleotide polymorphism of FABP4 gene and meat quality and expression of lipid metabolism genes in Yanbian yellow cattle

PLOS ONE

Dear Dr. Guangjun Xia,

Thank you for submitting your manuscript to PLOS ONE. After careful consideration, we feel that it has merit but does not fully meet PLOS ONE’s publication criteria as it currently stands. Therefore, we invite you to submit a revised version of the manuscript that addresses the points raised during the review process.

We would appreciate receiving your revised manuscript within the next 4 weeks. We urge you to recheck the statistical analyis of your study. To enhance the reproducibility of your results, we recommend that if applicable you deposit your laboratory protocols in protocols.io, where a protocol can be assigned its own identifier (DOI) such that it can be cited independently in the future. For instructions see: http://journals.plos.org/plosone/s/submission-guidelines#loc-laboratory-protocols

We look forward to receiving your revised manuscript.

Kind regards,

Heiner Niemann

Academic Editor

PLOS ONE

Additional Editor Comments (if provided):

The paper has been assessed by two expert reviewers. Both found the paper potentially interesting. However, a problem with the statistics was identified. Thus I urge the authors to contact an expert in statistics and have him/her check the solidity of the present analysis. Apart from that, the authors must address all points raised by the reviewers when they decide to revise the paper.

Journal Requirements:

The animal ethics committee approved the study.

Please amend your current ethics statement to include the full name of the ethics committee that approved your specific study.

For additional information about PLOS ONE submissions requirements for ethics oversight of animal work, please refer to http://journals.plos.org/plosone/s/submission-guidelines#loc-animal-research  

Reviewers' comments:

Reviewer's Responses to Questions

**Comments to the Author**

1. Is the manuscript technically sound, and do the data support the conclusions?

Reviewer #1: Partly

Reviewer #2: Yes

2. Has the statistical analysis been performed appropriately and rigorously? 

Reviewer #1: I Don't Know

Reviewer #2: I Don't Know

3. Have the authors made all data underlying the findings in their manuscript fully available?

Reviewer #1: No

Reviewer #2: Yes

4. Is the manuscript presented in an intelligible fashion and written in standard English?

Reviewer #1: No

Reviewer #2: Yes

5. Review Comments to the Author

Reviewer #1: In general, this is an interesting manuscript. It has a relatively limited audience as the breed of cattle utilized is small and regional. There is missing detail in the methods and in figure and table descriptions that must be added prior to publication.

Specific comments:

Line 64- Please add detail to ethics statement- specify institutional committee and protocol number.

Line 65- Add space after committee

Line 65- should be 30 months of age

Line 65- Specify slaughter condidtions and humane slaughter method

LIne 74 provide reference for Sox's method

Line 79- Specify manufactuers instructions and capitalize name of kit and full manufacturer

LIne 80- Provide method for DNA quantification

Table 1- Spacing in Table title is off with second line centered

Table 1- Annealing Temperature is spelled incorrectly and temperatures are missing for all but on of primer sets

Line 99- Correct capitalization in kit namee

Line 101- specify how RNA quality was assessed in gel

Line 115- 124- No detail is provided on how statistics were run and all methods need references

Figure 2- Need better description including what is in each lane and what ladder is

Figure 9- Image quality is poor and needs more detailed figure description

Table 2- Genotype is spelled wrong on table- also column heading need complete descriptions

Line 171: need reference and description of SHEsis software analysis

Table 3- Needs description of what is in Table

Figure 10- no informative and image quality is poor

Table 4- Column Heading have weird spacing and poorly done text. Need complete table description and superscripts are hard to read. Should also be referred to as superscript in table footnote

Line 197 to 202- Need description of gene relevance and function

Figure 11- use of color is unnecessary- letters are confusing and should be indicated in a different manner like * for p <0.05 and ** for p <0.01.

Table 5- Redudant and same comment as above

LIne 266 and 267- Parentheses are printing on top of numbers and letters in Acknowledgements

Reviewer #2: Dear Authors:

Very interesting and detailed assessments of FABP4 alleles in you native cattle. I see that animals were fed 30 month on a medium energy diet and once fattening is achieved even here you got marbling. I told the editor that I can not provide an expert review for your statistical-quantitative genetic analysis as I am a biochemist and not a population geneticist. Nevertheless as a whole this is commendable work; however your final conclusion about use of FABP4 as a marbling marker is certainly not new and marbling increases as cattle get fatter. You did however rule out roles for the alleles you assessed.

Intramuscular fat adipocyte size are not uniform in "muscle" tissue and neither are reletive expression of PPAR gamma, FAS, ACC expressions etc, so by not using some kind of microscopic dissection, how much fat and lean were in your tissue samples used for RNA isolation? I think this may have influenced lipogeneic gene expression data as there is not lipogenic activity in skeletal muscle cells and FABP4 usually reflects the use of exogenous fatty acid (from fat store turnover) as a source for im-fat accumulation. Under these circumstances the fattest animals had the most marbling as expected.

6. PLOS authors have the option to publish the peer review history of their article (what does this mean?). If published, this will include your full peer review and any attached files.

Reviewer #1: No

Reviewer #2: No

---

## [Author Response · Author response to Decision Letter 0]

13 May 2020

Reply to the editor:

1.We urge you to recheck the statistical analyis of your study. 

Response：According to your suggestion, we have asked a statistical expert to analyze the reliability of the data. According to the expert's suggestion, we have made the following modifications:

(1)According to the data, the SNP1 locus is in the hardy-Weinberg equilibrium state, which we have modified in the manuscript.

(2)we have added a reference for genetically-related indicators in the data analysis; the P value of chi-square test is also added in the comments after table 2.

(3)Although SNP1-6 was sequenced as well as only the SNP1 was found to be segmented by enzymes. As suggested, we have added additional notes in the manuscript.

2.Response: In this revision, we have not changed any financial disclosure issues. 

3.To enhance the reproducibility of your results, we recommend that if applicable you deposit your laboratory protocols in protocols.io, where a protocol can be assigned its own identifier (DOI) such that it can be cited independently in the future.

Response: We have uploaded the experimental protocol to protocols.io (https://dx.doi.org/10.17504/protocols.io.bf3zjqp6).

4.Additional Editor Comments (if provided):

The paper has been assessed by two expert reviewers. Both found the paper potentially interesting. However, a problem with the statistics was identified. Thus I urge the authors to contact an expert in statistics and have him/her check the solidity of the present analysis. Apart from that, the authors must address all points raised by the reviewers when they decide to revise the paper.

Response：According to your suggestion, we have asked a statistical expert to analyze the reliability of the data. According to the expert's suggestion, we have made the following modifications:

(1)According to the data, the SNP1 locus is in the hardy-Weinberg equilibrium state, which we have modified in the article.

(2) we have added a reference for genetically-related indicators in the data analysis; the P value of chi-square test is also added in the comments after table 2.

(3) Although SNP2-6 was sequenced as well as only the SNP1 was found to be segmented. As suggested, we have added additional notes in the manuscript.

Journal Requirements:

Response: We have changed the manuscript and file name strictly according to the template and style requirements of PLOS ONE.

2. Please amend your current ethics statement to include the full name of the ethics committee that approved your specific study.

Response: Thank you for your reminding, we have already improved the information related to our ethical declaration in the manuscript and the submission form.

3. PLOS ONE now requires that authors provide the original uncropped and unadjusted images underlying all blot or gel results reported in a submission’s figures or Supporting Information files. When you submit your revised manuscript, please ensure that your figures adhere fully to these guidelines and provide the original underlying images for all blot or gel data reported in your submission. 

Response: According to the requirements of PLOS ONE, we have collated and provided the original uncropped and unadjusted images in the manuscript. I posted at a public data repository (https://dx.doi.org/10.17504/protocols.io.bf3zjqp6).I have explained that in the cover letter.

Reviewer #1: In general, this is an interesting manuscript. It has a relatively limited audience as the breed of cattle utilized is small and regional. There is missing detail in the methods and in figure and table descriptions that must be added prior to publication.

Specific comments:

1.Line 64- Please add detail to ethics statement- specify institutional committee and protocol number.

Response: Thanks to the reviewer for the comment, we have added the details of the ethical statement.

2. Line 65- Add space after committee

Response: As suggested by the reviewer, we have corrected the same mistakes in the manuscript.

3. Line 65- should be 30 months of age

Response: We have modified it according to the suggestion of the reviewer.

4.Line 65- Specify slaughter condidtions and humane slaughter method

Response: We have supplemented the slaughter conditions and humane slaughter methods of experimental cattle in the materials and methods.

5. Line 74 provide reference for Sox's method

Response: We provide references and a brief description of the Sox's approach in accordance with the comments of the reviewers.

6. Line 79- Specify manufactuers instructions and capitalize name of kit and full manufacturer

Response: Thanks to the comments of the reviewers, we have corrected the spelling of the name of the kit and supplemented and improved the manufacturer's information. And we have added the instructions for using the kit in the supporting information.

7. Line 80- Provide method for DNA quantification

Response: As suggested by the reviewers, we have provided the method of DNA quantification in this paper.

8. Table 1- Spacing in Table title is off with second line centered

Response: We have checked and fixed the same errors in the text.

9. Table 1- Annealing Temperature is spelled incorrectly and temperatures are missing for all but on of primer sets

Response: Thanks to the reviewers for their careful review, we have corrected the spelling errors in the paper and added the missing annealing temperature in the table.

10. Line 99- Correct capitalization in kit name

Response: We have corrected the spelling errors of the kit in the text.

11. Line 101- specify how RNA quality was assessed in gel

Response: We have added how to evaluate the quality of RNA in gels.

12. Line 115- 124- No detail is provided on how statistics were run and all methods need references

Response: Thanks for the review. Our experimental cattles were of the same breed from the same cattle farm. They were slaughtered in the same time period. Therefore, one-way analysis of variance (ANOVA) was adopted. We have supplemented statistics and methods according to the comments.

13. Figure 2- Need better description including what is in each lane and what ladder is

Response: We have added a description of the gel electrophoresis pattern and indicated the specific name for each lane.

14. Figure 9- Image quality is poor and needs more detailed figure description

Response: We have modified the picture quality and added and improved the relevant information of the picture.

15. Table 2- Genotype is spelled wrong on table- also column heading need complete descriptions

Response: We have corrected spelling errors and supplemented the data in the table.

16. Line 171: need reference and description of SHEsis software analysis

Response: We have supplemented the reference and description of the SHEsis software analysis with a detailed analysis of the tables and diagrams.

17. Table 3- Needs description of what is in Table

Response: As mentioned above, we have supplemented the contents of the table.

18. Figure 10- no informative and image quality is poor

Response: Thank you for your comments. We have improved the quality of the pictures and supplemented them.

19. Table 4- Column Heading have weird spacing and poorly done text. Need complete table description and superscripts are hard to read. Should also be referred to as superscript in table footnote

Response: Thank you for your questions. We have modified and improved the table and added the description of the table.

20. Line 197 to 202- Need description of gene relevance and function

Response: Thanks for your reminding, we have added the function and correlation of several expressed genes in the paper.

21. Figure 11- use of color is unnecessary- letters are confusing and should be indicated in a different manner like * for p <0.05 and ** for p <0.01.

Response: Thanks for your suggestion. We have modified the bar chart to change the representation of difference significance to make it more intuitive.

22. Table 5- Redudant and same comment as above

Response: Thank you very much for your suggestion. We did not modify table 5, because in the third row of table 5, there is no significant difference between wild-type individuals and heterozygous individuals, and there is significant difference between wild-type individuals and mutant individuals, while there is no significant difference between heterozygous individuals and mutant individuals. Using * does not intuitively reflect the relationship between the three.

23. Line 266 and 267- Parentheses are printing on top of numbers and letters in Acknowledgements

Response: Thank you for your careful review. We have adjusted and modified the contents of the acknowledgements according to the requirements of PLOS ONE.

Reviewer #2: Dear Authors:

Very interesting and detailed assessments of FABP4 alleles in you native cattle. I see that animals were fed 30 month on a medium energy diet and once fattening is achieved even here you got marbling. I told the editor that I can not provide an expert review for your statistical-quantitative genetic analysis as I am a biochemist and not a population geneticist. Nevertheless as a whole this is commendable work; however your final conclusion about use of FABP4 as a marbling marker is certainly not new and marbling increases as cattle get fatter. You did however rule out roles for the alleles you assessed.

 Response: I quite agree with the reviewer. There are many reports on FABP4 as marbling marker. However, no research has been done on Yanbian yellow cattle of five famous local superior breeds in China. This study is helpful to elucidate the genetic effect of FABP4 gene in Yanbian yellow cattle population, and to provide reference for molecular assisted breeding of meat quality traits of this local breed.

Intramuscular fat adipocyte size are not uniform in "muscle" tissue and neither are relative expression of PPAR gamma, FAS, ACC expressions etc, so by not using some kind of microscopic dissection, how much fat and lean were in your tissue samples used for RNA isolation? I think this may have influenced lipogeneic gene expression data as there is not lipogenic activity in skeletal muscle cells and FABP4 usually reflects the use of exogenous fatty acid (from fat store turnover) as a source for im-fat accumulation. Under these circumstances the fattest animals had the most marbling as expected.

Response: At present, it is very difficult to micropartition intramuscular fat cells. The sample processing method in this study is commonly used in current studies on intramuscular fat metabolism genes (including RNA-seq), it may have an impact on the results, but a reliable conclusion can be drawn based on previous research.

---

## [Editor Report · Decision Letter 1]

14 May 2020

PONE-D-20-01041R1

Correlation between single nucleotide polymorphism of FABP4  gene and meat quality and expression of lipid metabolism genes in Yanbian yellow cattle

PLOS ONE

Dear Dr. Xia,

Thank you for submitting your manuscript to PLOS ONE. After careful consideration, we feel that it has merit but does not yet fully meet PLOS ONE’s publication criteria as it currently stands. Therefore, we invite you to submit a revised version of the manuscript that addresses the points raised during the review process.

Please make sure that the paper adheres to English grammar throughout the text.

We would appreciate receiving your revised manuscript by May 24th, 2020. To enhance the reproducibility of your results, we recommend that if applicable you deposit your laboratory protocols in protocols.io, where a protocol can be assigned its own identifier (DOI) such that it can be cited independently in the future. For instructions see: http://journals.plos.org/plosone/s/submission-guidelines#loc-laboratory-protocols

We look forward to receiving your revised manuscript.

Kind regards,

Heiner Niemann

Academic Editor

PLOS ONE

Additional Editor Comments (if provided):

The authors have dealt adequately with the points raised by the reviewers which improved the quality of the paper. One problem remains: There are several errors of the English grammar which to be eliminated prior to acceptance for publication. I urge the authors to contact a native English speaker.

---

## [Author Response · Author response to Decision Letter 1]

21 May 2020

1. Please make sure that the paper adheres to English grammar throughout the text.

 Response: We have asked Editage to get a native English speaker to revise the entire manuscript. 

2. To enhance the reproducibility of your results, we recommend that if applicable you deposit your laboratory protocols in protocols.io, where a protocol can be assigned its own identifier (DOI) such that it can be cited independently in the future. 

 Response: We have updated the protocol of our experiment according to the newly revised manuscript. The DOI address is dx.doi.org/10.17504/protocols.io.bgnjjvcn.

3. There are several errors of the English grammar which to be eliminated prior to acceptance for publication. I urge the authors to contact a native English speaker.

 Response: We have asked Editage to get a native English speaker to revise the entire manuscript. 

4. While revising your submission, please upload your figure files to the Preflight Analysis and Conversion Engine (PACE) digital diagnostic tool. 

 Response: As requested by your magazine, we have uploaded the graphics file to PACE.

---

## [Editor Report · Decision Letter 2]

26 May 2020

Correlations between single nucleotide polymorphisms in FABP4 and meat quality and lipid metabolism gene expression in Yanbian yellow cattle

PONE-D-20-01041R2

Dear Dr. Xia,

We are pleased to inform you that your manuscript has been judged scientifically suitable for publication and will be formally accepted for publication once it complies with all outstanding technical requirements.

With kind regards,

Heiner Niemann

Academic Editor

PLOS ONE

Additional Editor Comments (optional):

The authors have dealt with the suggestion of improving English grammar and language. The paper is now ready to be accepted for publication.
---

## [Editor Report · Acceptance letter]

29 May 2020

PONE-D-20-01041R2 

Correlations between single nucleotide polymorphisms in *FABP4* and meat quality and lipid metabolism gene expression in Yanbian yellow cattle 

Dear Dr. Xia:

I am pleased to inform you that your manuscript has been deemed suitable for publication in PLOS ONE. Congratulations! Your manuscript is now with our production department. 

With kind regards,

on behalf of

Dr. Heiner Niemann 

Academic Editor

PLOS ONE